# Depressive Symptoms and Migrant Worker Wages: Estimation Based on a Nationally-Representative Panel Dataset

**DOI:** 10.3390/ijerph16061009

**Published:** 2019-03-20

**Authors:** Li Huang, Xue Zhang, Mi Zhou, Brendan Nuse, Liuyin Tong

**Affiliations:** 1College of Economics and Management, Shenyang Agricultural University, Shenyang 110866, Liaoning, China; lilyhuang1983013@163.com (L.H.); zhangxuesn@163.com (X.Z.); tongliuyin@163.com (L.T.); 2College of Liberal Arts, Shanxi Agricultural University, Jinzhong 030801, Shanxi, China; bnuse@oberlin.edu

**Keywords:** migrant workers, depressive symptoms, wage, working stability, Coarsened Exact Matching method

## Abstract

In recent years, migrant workers, defined as people who move from Chinese rural areas to cities in other parts of the country to find work, have experienced slowed wage growth. An important question that has emerged is whether depressive symptoms have a significant relationship with migrant worker wages. This paper uses a nationally representative panel dataset to examine the overall association of depressive symptoms and migrant worker wages in China and explores the indirect mechanisms through which these impacts occur. Using the Coarsened Exact Matching method, our results show that depressive symptoms have a significant direct negative relationship with migrant worker wages, and that this relationship is consistent. Furthermore, we also find that depressive symptoms can reduce migrant worker wages indirectly by increasing the frequency of job conversion or by shortening work duration.

## 1. Introduction

In recent years, the total number of migrant workers, which, in this paper, refers to workers who move from Chinese rural areas to cities in other parts of China to find work, has increased continuously. In 2017, according to the “Statistical Bulletin on the Development of Human Resources and Social Security in 2017” published by the Ministry of Human Resources and Social Security of the People’s Republic of China, the total number of migrant workers nationwide was approximately 286 million, an increase of 4.81 million over the previous year. However, there is still a gap between migrant and urban worker wages. In fact, the wage growth rate of migrant workers has fallen sharply in recent years. According to the National Bureau of Statistics, the migrant worker wage growth fell from 13.9% in 2013 to 6.4% in 2017, compared with 10% for urban workers and slower than even the GDP growth rate of 6.9% that year. This means that the wage gap between migrant workers and urban workers will become wider and wider, which will not only aggravate social conflicts, but also hinder the integration of migrant workers into cities, and ultimately prevent China from successfully crossing the middle-income trap.

Studies in China and other developing countries have identified that education level, discrimination, skill level, work experience, and health status affect migrant worker wages [1,2,3,4]. However, much less attention has been paid to the role of mental illness on migrant worker income level in the context of developing countries (such as China). In fact, since Heckman et al. [5] first included the non-cognitive skill factor in the wage determination equation, the impact of emotion on wages has been widely considered by economists. Studies in developed countries have shown that emotional stability may have a significant impact on income [6,7,8], or no significant impact, or only have a significant impact on women [9]. The literature already available suggests that negative psychological factors in young Americans have a negative impact on their future income (in this paper, negative emotions can be described as any feeling which causes you to be miserable and sad. These emotions make you dislike yourself and others, and take away your confidence. Negative impact on wages means a reduction in earnings. And many of the negative psychological risk factors linked to dementia are associated with stress—or, more precisely, distress) [10]. We are interested in whether the negative emotions of migrant workers have an impact on their wages. More critically, according to estimates, the prevalence of depressive symptoms of migrant workers has reached 23.7% in China [11], which means that nearly 60 million of China’s nearly 300 million migrant workers suffer from depression.

However, research on these issues in developing countries is still limited. Although there have been a number of studies examining the association between emotions and wages in China (see for example, [12,13,14,15]), such studies almost always focus on positive emotions rather than negative ones. To our knowledge, there have been no studies on the impact of negative emotions such as depressive symptoms on migrant worker wages in China.

The overall goal of this study is to examine the general relationship between depressive symptoms and migrant worker wages in China. In pursing this research goal, we examine the overall association between depressive symptoms and migrant worker wages by using a nationally representative panel dataset and explore the indirect mechanisms through which this impact occurs. Specifically, these mechanisms include increased frequency of job conversion and decreased work duration. To perform this study, this paper uses a Coarsened Exact Matching (CEM) model, which can improve sample matching.

The rest of the paper is organized as follows: Section 2 describes the sampling methods, data collection and the way we measure depressive symptoms. Section 3 introduces the theoretical framework and model. Section 4 presents our results of the relationships between depressive symptoms and migrant worker wages. It also examines the mechanisms of the association between depressive symptoms and migrant worker wages (namely, frequency of job conversion and decreased work duration). Section 5 is a conclusion.

## 2. Data

### 2.1. Data Source and Sampling

The data used for this study come from the China Family Panel Studies (CFPS) for the years 2012 and 2014. The CFPS is a nationally representative, longitudinal social survey that was launched in 2010 and is conducted biennially by the Institute of Social Science Survey (ISSS) at Peking University, China. The survey design is based on the Panel Survey of Income Dynamics (PSID), the National Longitudinal Surveys of Youth (NLSY), and the Health and Retirement Study (HRS) in the United States. It focuses on a range of topics related to educational outcomes, economic activities, migration, health, and family dynamics. The survey collects data at three levels: the individual, family, and community levels.

The CFPS surveyed respondents in sampling units in 25 provinces (all provinces except Xinjiang, Tibet, Qinghai, Inner Mongolia, Ningxia, and Hainan), a sampling frame which represents 95% of the Chinese population. To create a nationally and provincially representative sample, the CFPS adopted a “Probability-Proportional-to-Size” (PPS) sampling strategy with multi-stage stratification and carried out a three-stage sampling process [16,17]. The first stage was the Primary Sampling Unit, in which county-level units were randomly selected. In the second stage, village-level units (referring to villages in rural areas and neighborhoods/communities in urban areas) were selected. In the third stage, households from the village-level units were selected according to the study’s systematic sampling protocol. Enumerators interviewed all members of each household who were home at the time of the survey.

For the purposes of the survey, we examined the case of the adult sample of CFPS in 2012 and 2014, and created a panel dataset that includes 1686 migrant workers who worked wage jobs during both years. The dataset excludes observations with missing information (Figure 1).

### 2.2. Measures

The dependent variable in this study is migrant worker wages. The depression scale used for the 2012 CFPS data is the Center for Epidemiologic Studies Depression Scale (CES-D). The CES-D scale has been used widely in the international literature to evaluate depressive symptoms [18,19]. As shown in Table A1 in the Appendix A, the CES-D includes 20 items and is scored on a Likert scale with four possible answers corresponding to how often the respondents experienced a given emotion within the past week: “rarely or none of the time (less than 1 day)”, “some or a little of the time (1–2 days)”, “occasionally or a moderate amount of time (3–4 days)”, and “most or all of the time (5–7 days)”. Possible scores range from 0–60 and a score of 16 or higher is indicative of depression. The Chinese version of CES-D has been used in previous research [20], and its reliability and validity has been tested among Chinese populations [21]. Therefore, the scale has been confirmed to be appropriate for use in China [22].

We used two indicators “Did subjects change jobs between 2012 and 2014?” and “Work duration” to measure migrant worker job stability. In addition, there were a number of smaller blocks that were used to enumerate migrant worker and family characteristics and other control variables. Specifically, the data allow us to generate variables that measure personal characteristics, including gender, age, education, minority status, marital status, etc. Descriptive statistics of the variables used in this paper are shown in Table 1.

## 3. Methodology

### 3.1. Ordinary Least Squares Regression

Ordinary least-squares (OLS) regression is a generalized linear modelling technique that may be used to model a single response variable which has been recorded on at least an interval scale. The technique may be applied to single or multiple explanatory variables and also categorical explanatory variables that have been appropriately coded [23].

Ordinary Least squares Regression is represented as follows:(1)$$Y=\alpha +\beta_{1}X_{1}+\beta_{2}X_{2}+\varepsilon$$
where Y is the outcome variable, in this paper this refers to the wage, *X*_1_ is a dummy variable to indicate whether the individual has depressive symptoms or not, *X*_2_ represents other control variables, including gender, age, education level, marital status, and minority status, β_1_, and β_2_ are regression coefficients and ε is the random error term.

### 3.2. Coarsened Exact Matching (CEM)

There may be differences in personal characteristics between those migrant workers who show depressive symptoms and those who do not. For example, women [24] and poorly educated laborers [25] are more likely to suffer from depression. These characteristics may also affect their wages, which makes simple comparison of the wage differences between migrant workers with and without depressive symptoms difficult, due to endogenous problems.

Therefore, in this paper, the migrant workers with and without depressive symptoms in 2012 were regarded as the treatment group and control group, respectively. The heterogeneity between the two groups was eliminated using the Coarsened Exact Matching method, and individuals who matched successfully between groups were retained. Then, we estimated the endogeneity of controlling depressive symptoms by the disposal effect, and identify the wage difference between migrant workers with depressive symptoms and those without symptoms.

CEM is a monotonoic imbalance reducing matching method, which means that the maximum imbalance between the treatment group and the control group can be selected by the user in advance. Adjusting the imbalance on one variable does not affect the imbalance of any other variable.

For a specific migrant worker, *Inc*_1_*_i_* refers to the wage in 2014 for workers who had depressive symptoms in 2012, *Inc*_0_*_i_* refers to the wage in 2014 for workers who did not have depressive symptoms in 2012. *Inc*_0_*_i_* and *Inc*_1_*_i_* are the potential result of the individual in the treatment group and control group respectively. *X_i_* is the individual characteristic vector influencing *Inc_i_*. The results equation for the two groups is as follows:{(2)Inc0i=β0Xi+ε0i if Depi=0 (3)Inc1i=β1Xi+ε1i if Depi=1

Equation (2) is applicable to individuals without depressive symptoms in 2012, and Equation (3) is applicable to individuals with depressive symptoms in 2012. W_i_ is the characteristic vector affecting the existence of depressive symptoms in individual i, and the selection equation is shown as follows:(4)$$Dep_{i}=\{\begin{array}{c} 1\;\mathrm{if}\;\alpha W_{i}+\eta_{i}>0\\ 0\;\mathrm{others}\; \end{array}$$

If there is a systematic difference in the vector W_i_ between the treatment group and the control group, the presence of depressive symptoms is not the result of random selection, and the endogeneity of *Dep_i_* will create sample selection bias when *X_i_* and *W_i_* cross. The disposition effect is estimated by controlling for individual characteristics that affect both *Dep_i_* and *Inc_i_*. The mean value of potential Inc_0i_ in the treatment group is independent of the *Dep*_i_ conditions. The potential result of the treatment group is simulated by adjusted Inc_0i_ of the control group to estimate the average treatment effect of the treatment group:(5)$$\mathrm{ATET}=E\left({\mathrm{Inc}}_{1}-{\mathrm{Inc}}_{0}{|\mathrm{Dep}}_{i}=1\right)$$

## 4. Results

### 4.1. The Relationships between Depressive Symptoms and Migrant Worker Wage Earnings

Our analysis using ordinary least squares (OLS) regression analysis shows that migrant worker depressive symptoms in 2012 had a significant negative correlation with migrant worker wages in the same year, and also on their wages in 2014. (Table 2). Specifically, when we examine the unrestricted model in Column 1, we find that depressed migrant workers earned 29% less in 2012 than non-depressed workers (significant at the 1% level—Row 1, Column 1) and we find similar correlations in the restricted model in Column 2 (19%—significant at the 1% level). Additionally, the unrestricted model in Column 3 shows that migrant workers with depressive symptoms in 2012 earned 22% less in 2014 than non-depressed workers (significant at the 1% level), as well as in the restricted model in Column 4 (14%—significant at the 1% level).

In order to eliminate OLS model estimation bias, we further used the CEM method to measure the difference in migrant worker wages in 2014 between those who exhibited depressive symptoms in 2012 and those who did not in the case of matching the characteristic variables (including gender, age, marital status, education, minority and region, etc.) that affect migrant worker wages between the treatment group (migrant workers group with depressive symptoms in 2012) and the control group (migrant workers group without depressive symptoms in 2012). The matching result in Table 3 shows that the value of τ_1_ after matching is significantly lower than the value before matching, which effectively reduces the heterogeneity between the two groups.

Table 4 reports the sample average treatment effect of migrant workers’ depressive symptoms in 2012 on their wages in 2014. It can be seen that the depressive symptoms in 2012 caused a wage reduction of about 2300 yuan in 2014 (significant at the 5% level), which is equivalent to the average monthly salary of the sample used in this study in 2014. Overall, the results obtained by the CEM method are consistent with those of the OLS model.

### 4.2. The Mechanism of the Association between Depressive Symptoms and Migrant Worker Wages

This paper further examines and identifies the mechanism of the association between depressive symptoms and migrant worker wages from the perspectives of job conversion and work duration.

According to the statistics in Figure 2, 31.2% of the migrant workers with depressive symptoms in 2012 had a job conversion during the period between the two surveys, whereas only 25.2% of the workers without depressive symptoms had one. There was a difference in the significance level of 5% between the two groups of migrant workers who had job conversion, which indicated that migrant workers with depressive symptoms were more prone to switch jobs and lacked job stability.

Further, the estimated results of the Probit regression model (Table 5) show that whether or not control variables (including gender, age, education, marital status, minority status and region) were included, the association between depressive symptoms and migrant worker job conversion was significantly positive, which indicated that migrant workers with depressive symptoms in 2012 had a higher frequency of job conversion than those without depressive symptoms. Frequent job conversion will lead to the loss of specialized human capital and significantly reduce migrant worker wage level [26,27]. Depressive symptoms, therefore, can reduce migrant worker wages by increasing the frequency of job turnover.

Additionally, if migrant workers switch jobs too frequently due to depressive symptoms, this frequent job turnover will have a negative impact on their wages. As shown in Figure 3, the work duration of the 646 subjects without depressive symptoms in 2012 averaged 20 months, while that of the 273 subjects with depressive symptoms in 2012 averaged 19 months. At the same time, there was a 10% significant difference in the work duration between the two groups, which indicates that depressive symptoms may shorten the duration of the same job.

Furthermore, the estimated results of the OLS regression model (Table 5) show that depressive symptoms negatively affected the working duration of migrant workers at a significant level of 10%, regardless of whether or not control variables are added. Depressive symptoms, therefore, may reduce migrant worker wages by shortening the duration of a certain job.

## 5. Discussion

Using nationally representative data from the CFPS, we demonstrate that the prevalence of migrant worker depressive symptoms in China is high. Specifically, according to the CES-D scale, 29% of all migrant workers in China display symptoms of depression. These rates are high compared to similar studies that use nationally-representative samples from other developing countries [28]. These estimates also support previous studies in China that reported a high prevalence of migrant worker depression—although previous analyses used data from studies that were based on much smaller samples or from more limited geographic areas [29,30]. Taken together, these findings indicate that migrant workers’ depressive symptoms are indeed an issue in China, and steps must be taken to improve the mental health of Chinese migrant workers.

Our results also show that migrant workers with depressive symptoms were more prone to switch jobs and lacked job stability. In addition, we discovered that migrant workers with depressive symptoms stayed at the same job for a shorter amount of time than those without depressive symptoms did. As many studies have shown, the unemployment rate and probability of unemployment are significantly higher for those with depressive symptoms than for those without depressive symptoms [31,32,33,34]. At the same time, depression may also prolong the gap period between jobs [35]. These findings are important because they have allowed us to identify the mechanism determination of migrant worker wages in China. With this knowledge, it will be possible to develop policies and programs targeting these migrant workers’ depressive symptoms that can increase their productivity across China.

Although no previous studies have comprehensively examined the relationship between depressive symptoms and migrant worker wages in China, our findings are supported by the international literature. For example, a study conducted in the United States found that depressive symptoms are associated with subsequent unemployment and loss of family income among working young adults [36]. In addition, another study from the United States found that major depression was associated with $2,838 lower personal income [37]. Findings from other countries on the relationship between depressive symptoms and income also support the findings of our research. Research conducted in India has found that mental health was positively correlated with higher income level [38]. It has also been found in a Spanish sample that migrant workers whose monthly income decreased suffered an increased risk of poor mental health [39].

We use panel data to test whether depressive symptoms are associated with wage. But there are 927 individuals missing from 2012 to 2014, accounting for 24.66% of the total. One possible explanation for this loss is that migrant workers are highly mobile and work shifts are too frequent during the two periods, making it difficult to track subjects. Meanwhile, a sample of 1146 migrant workers who stopped working as wage earners in 2014 was removed from the sample. This sample had accounted for 40.5 percent of the workers tracked in 2014. In order to test whether there are systematic differences between individuals in the lost sample as well as between individuals in the further deleted samples and those who do not fall into these two categories, we have added a balance test using the Probit model. As Table A2 in Appendix A shows, we generally did not find systematic differences between them.

Besides the missing individuals, the years considered in the research (2012–2014) are the years of the world economic crisis, which may affect the likelihood of migrant workers perceiving insufficient individual work demand [40,41,42,43] and experiencing salary reductions [44,45,46], which increase the presence of depressive symptoms. However, in China, the economic crisis was not as significant as in other parts of the world, especially for migrant workers from rural areas, because migrant workers can return to rural areas if they cannot find a suitable job in an urban area [47].

This study has a number of strengths. First, our sampling frame represents 95% of the Chinese population, and, therefore, can be considered nationally representative. Second, the large sample size of our panel data-set (*n* = 1686) gives our research a high degree of statistical power and considerable external validity. Third, the Institute of Social Science Survey (ISSS) of Peking University collected all data using a common sampling strategy. Last, this paper focuses on the relationship between depressive symptoms and migrant workers not only on their wage, but also on their job conversion and work duration. Some studies recognized that migrant workers are more likely to have occupational injuries [48,49,50,51] and may cause long absences and difficulty returning to work [52] which may induce a worsening of depressive symptoms. However, the conversion of occupation is one of the main factors recognized by WHO to reduce the burden of depression [53]. In this paper, we analyze the relationship between job conversion and wage decline.

Despite its strengths, our study also suffers from several limitations. First, the CFPS only collected self-reported information, which limited us to examining migrant workers’ depressive symptoms with medical evaluation. Second, because a person losing his or her job or earning less money probably is more likely to report depressive symptoms when interviewed with a survey, our research only indicates the association between migrant workers’ depressive symptoms and their income. Future research examining the causal effect of migrant workers’ depressive symptoms on their income in China would benefit from focusing on these research areas.

From a policy perspective, our results suggest that the universality of depressive symptoms in migrant workers must not be ignored. The Chinese government should publicize the existence of depressive symptoms and the harm they cause to migrant workers through various channels, so as to help them recognize their own potential depressive symptoms and lay a foundation for them to take further measures to address them. Relevant departments should fully consider the impact of unskilled human capital on policy effectiveness when formulating policies to improve human capital and achieve China’s smooth crossing of the “middle income trap”. We believe our results can help policy makers develop targeted programs for depressed migrant workers in the future.

## 6. Conclusions

Using nationally representative data, this study explored the relationship between depressive symptoms and migrant worker wages. In addition, we examined the mechanism between depressive symptoms and migrant worker wages from the perspectives of job conversion and work duration. Our results showed that nearly 30% of migrant workers exhibited depressive symptoms, which had a significant direct negative consistent relationship with migrant worker wages. Furthermore, depressive symptoms had a significant relationship with migrant worker wages through job stability. In particular, depressive symptoms can reduce migrant worker wages by increasing the frequency of job conversion or shortening the duration of the work.

## Figures and Tables

**Figure 1 ijerph-16-01009-f001:**
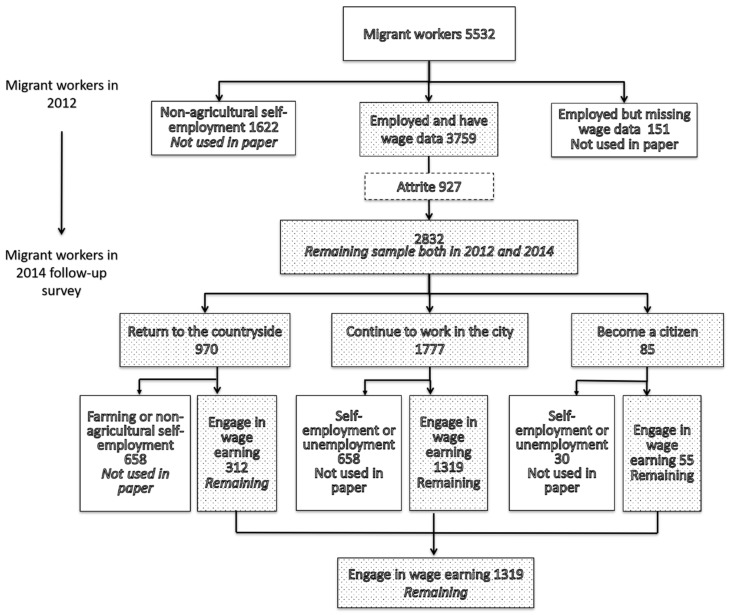
The construction of panel data. Source: China Family Panel Studies (CFPS) (2010, 2014).

**Figure 2 ijerph-16-01009-f002:**
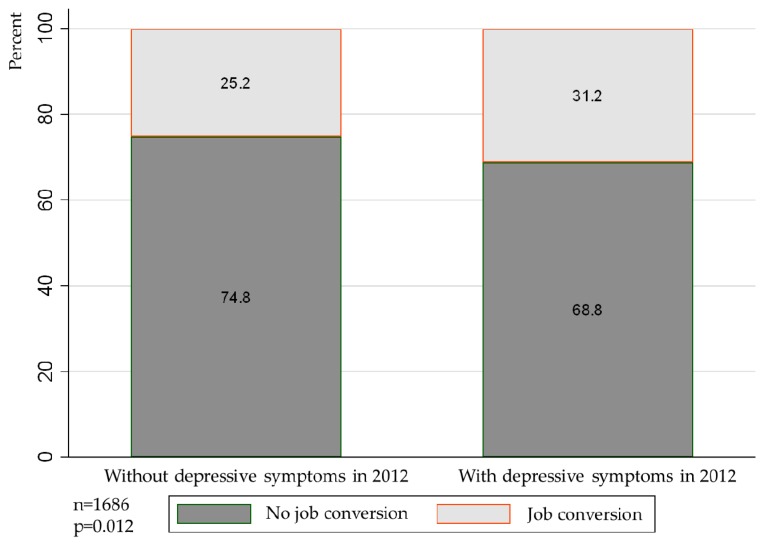
Depressive symptoms and job transition in 2012. Sources: CFPS (2012, 2014).

**Figure 3 ijerph-16-01009-f003:**
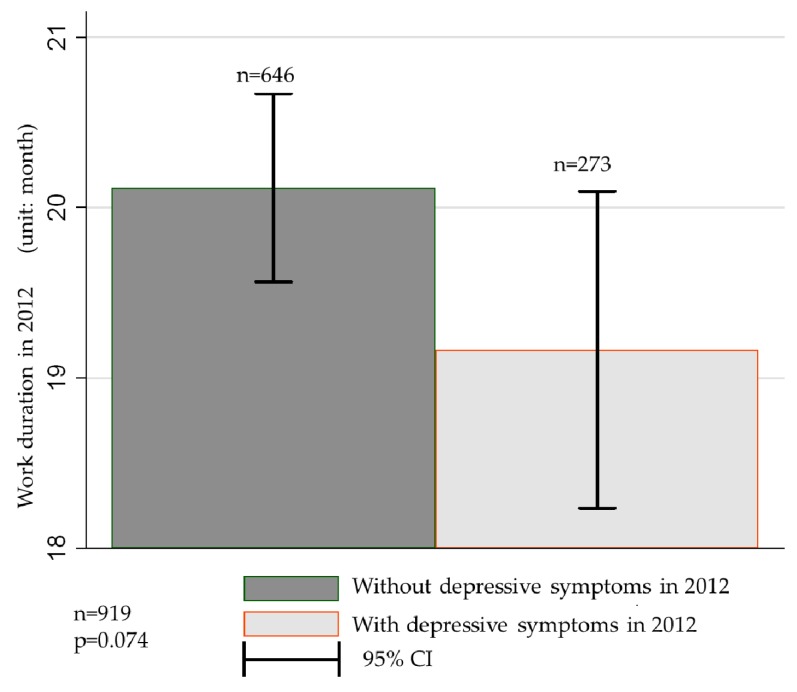
Depressive symptoms and work duration in 2012. Sources: CFPS (2012, 2014).

**Table 1 ijerph-16-01009-t001:** Descriptive statistics of variables.

Variables	Observations	Mean	SD	Min	Max
Wage earnings in 2012 (yuan)	1686	21,093.04	14,236.94	79	80,000
Wage earnings in 2014 (yuan)	1686	27,546.04	17,674.28	9	85,000
Depressive symptoms in 2012 (1 = yes, 0 = no)	1686	0.29	0.45	0	1
Job conversions between 2012 and 2014 (1 = yes, 0 = no)	1686	0.27	0.44	0	1
Work duration in 2012 (months)	919 ^a^	19.83	7.38	0	33
Age in 2012	1686	35.94	10.66	18	60
Gender (1 = Male, 0 = Female)	1686	0.70	0.46	0	1
Minority (1 = Han Chinese, 0 = non-Han Chinese)	1682 ^b^	0.95	0.22	0	1
Marital status in 2012 (1 = Married, 0= others)	1686	0.81	0.40	0	1
Marital status in 2014 (1 = Married, 0= others)	1686	0.85	0.36	0	1
Completion of nine-year compulsory education in 2012 (1 = yes, 0 = no)	1686	0.23	0.42	0	1
East area in 2012 (1 = yes, 0= others)	1686	0.45	0.50	0	1
Midland in 2012 (1 = yes, 0= others)	1686	0.34	0.47	0	1
East area in 2014 (1 = yes, 0= others)	1686	0.33	0.47	0	1
Midland in 2014 (1 = yes, 0= others)	1686	0.47	0.50	0	1

Sources: CFPS (2012, 2014); Note: ^a^ The missing value comes from the fact that there was no work at the time of the survey in 2012, or the work duration data was missing at the time of the survey in 2014. ^b^ Minority data has four missing values.

**Table 2 ijerph-16-01009-t002:** Ordinary Least Squares estimates of the relationships between migrant worker depressive symptoms in 2012 and their wages in 2012 and 2014.

Dependent Variable: Logarithm of Annual Wage	Without Control Variables and Fixed Effect	With Control Variables and Fixed Effect	Without Control Variables and Fixed Effect	With Control Variables and Fixed Effect
	2012	2012	2014	2014
Depressive symptoms in 2012 (1 = yes, 0 = no)	−0.29 *** (0.05)	−0.19 *** (0.05)	−0.22 *** (0.06)	−0.14 *** (0.05)
Constant	9.75 *** (0.03)	9.72 *** (0.14)	10.00 *** (0.03)	9.99 *** (0.10)
Control variables	no	yes	no	yes
Province fixed effect	no	yes	no	yes
R-squared	0.02	0.09	0.01	0.06
Observations	1686	1682	1686	1686

Sources: CFPS (2012, 2014); Note: The symbol *** means that the *p*-value is less than 0.01 (*p* < 0.01). The control variables added in the two-stage regression analysis, including gender, age, education level (whether or not the subject completed the nine-year compulsory education), marital status, and minority variables, are variables generated by each period of data (except the same variable used in the two stages of “minority”).

**Table 3 ijerph-16-01009-t003:** Results of analysis conducted using Coarsened Exact Matching.

	Full Sample		No Job Conversion		Job Conversion	
	No Depressive Symptoms	Depressive Symptoms	No Depressive Symptoms	Depressive Symptoms	No Depressive Symptoms	Depressive Symptoms
Number of successful matched samples	1044	447	752	297	210	117
Number of samples that failed to match	158	37	147	36	93	34
$\tau_{1}\left(\mathrm{Before}\;\mathrm{matching}\right)$	0.3777		0.3863		0.4250	
$\tau_{1}\left(\mathrm{After}\;\mathrm{matching}\right)$	0.1309		0.0001		0.1828	

Sources: CFPS (2012, 2014).

**Table 4 ijerph-16-01009-t004:** Sample average treatment effect of migrant worker depressive symptoms in 2012 on their wages in 2014.

Dependent Variable: Wage in 2014 (yuan)	Coefficient	SE	T Value	*p* Value
Depressive symptoms in 2012 (1 = yes, 0 = no)	−2311.02 **	981.23	−2.36	0.02
Constant	27,118.77 ***	537.26	50.48	0.00

Sources: CFPS (2012, 2014); Note: *** *p* < 0.01, ** *p* < 0.05.

**Table 5 ijerph-16-01009-t005:** Relationship between depressive symptoms and employment stability in 2012.

Dependent Variable	Were There Any Job Transitions during the Two Years? (1 = yes) Probit Model		Duration of Work (Unit: Month) OLS Model	
	Without Control Variables	With Control Variables	Without Control Variables	With Control Variables
Depressive symptoms in 2012 (1 = yes, 0 = no)	0.18 **	0.16 **	−0.95 *	−0.98 *
	(0.07)	(0.08)	(0.55)	(0.56)
Constant	−0.67 ***	0.59 **	20.12 ***	16.26 ***
	(0.04)	(0.24)	(0.28)	(2.01)
Control variables	no	yes	no	yes
Observations	1686	1659	919	901

Sources: CFPS (2012, 2014). Note: *** *p* < 0.01, ** *p* < 0.05, * *p* < 0.1, Robust standard errors in parentheses; The control variables were gender, age, education level (whether the respondents completed the nine-year compulsory education), marital status, whether they were Han Chinese and region (eastern or central regions) in 2014.

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
