# Peer review of "Depressive Symptoms and Migrant Worker Wages: Estimation Based on a Nationally-Representative Panel Dataset"

_ijerph, 2019, doi:10.3390/ijerph16061009_

Round 1
Reviewer 1 Report
The authors use a nationally representative panel dataset to examine the effects of depressive symptoms on migrant worker wages in China and explore the indirect mechanisms through which these impacts occur. The topic of this study is interesting and relevant to the special issue (Health Economics) of the International Journal of Environmental Research and Public Health. It is well written. I offer few comments to tighten the manuscript in preparation for publication.
A discussion section is needed. Please indicate similarities and differences between the results and the work of others. Also please state the importance of this study. What are the strengths of the study?
Author Response
Dear Qu:
Thank you very much for your letter on March 2, 2019. We want to thank you and the reviewer for your comments and suggestions on our manuscript, “Depressive Symptoms and Migrant Worker Wages: Estimation Based on a Nationally-Representative Panel Dataset” by Li Huang, Xue Zhang, Mi Zhou ,Brendan Nuse and Liuyin Tong. We believe that the comments were helpful and have helped greatly improve the manuscript. Hence, we are sending you, as requested, the revised version of our manuscript.
In making our revisions we have responded to all suggestions by you and the reviewer. Generally, we made the following major revisions:
1.) We have added a discussion section. In this part, we discussed the main results, the interesting points, the strengths and limitations of our study, and the policy implications of the study. We also compared our results with the results of other studies.
2.) We have changed the methodology section. We moved the mechanism to the discussion section. And we added an introduction to the OLS method.
We have included a detailed, point-by-point response to the reviewer in this letter. We are also providing a revised version of the manuscript in a Word document. We have tried to follow general IJERPH style, but, of course, stand ready to make any further changes that are needed (editorial or otherwise).
If you need anything else, please do not hesitate to contact me.
Sincerely,
Mi Zhou, Li Huang
Depressive Symptoms and Migrant Worker Wages: Estimation Based on a Nationally-Representative Panel Dataset
Response to Reviewer1:
First, we want to thank the reviewer for carefully reading our paper and providing useful, detailed comments – we believe addressing these comments has strengthened the paper considerably. We address each of the reviewer’s specific comments below. The original comments of the reviewer are in italics. Our responses are indented.
Comment 1: A discussion section is needed. Please indicate similarities and differences between the results and the work of others. Also please state the importance of this study. What are the strengths of the study?
Response to Comment1: Thank you very much for the suggestion. In the revised manuscript, we have added a discussion section. The revised discussion is now presented as follows (lines 233-304, pages 8-9):
“5. Discussion
Using nationally representative data from the CFPS, we demonstrate that the prevalence of internal migrant worker depressive symptoms in China is high. Specifically, according to the CES-D scale, 29% of all internal migrant workers in China display symptoms of depression. These rates are high compared to similar studies that use nationally-representative samples from other developing countries [28]. These estimates also support previous studies in China that reported a high prevalence of internal migrant worker depression—although previous analyses used data from studies that were based on much smaller samples or from more limited geographic areas [29-30]. Taken together, these findings indicate that internal migrant workers’ depressive symptoms are indeed an issue in China, and steps must be taken to improve the mental health of Chinese internal migrant workers.
Our results also show that internal migrant workers with depressive symptoms were more prone to switch jobs and lacked job stability. In addition, we discovered that internal migrant workers with depressive symptoms stayed at the same job for a shorter amount of time than those without depressive symptoms did. As many studies have shown, the unemployment rate and probability of unemployment are significantly higher for those with depressive symptoms than for those without depressive symptoms [31-34]. At the same time, depression may also prolong the gap period between jobs [35]. These findings are important because they have allowed us to identify the mechanism determination of internal migrant worker wages in China. With this knowledge, it will be possible to develop policies and programs targeting these internal migrant workers’ depressive symptoms that can increase their productivity across China.
Although no previous studies have comprehensively examined the relationship between depressive symptoms and internal migrant worker wages in China, our findings are supported by the international literature. For example, a study conducted in the United States found that depressive symptoms are associated with subsequent unemployment and loss of family income among working young adults [36]. In addition, another study from the United States found that major depression was associated with $2,838 lower personal income [37]. Findings from other countries on the relationship between depressive symptoms and income also support the findings of our research. Research conducted in India has found that mental health was positively correlated with higher income level [38]. It has also been found in a Spanish sample that migrant workers whose monthly income decreased suffered an increased risk of poor mental health [39].
We use panel data to test whether depressive symptoms are associated with wage. But there are 927 individuals missing from 2012 to 2014, accounting for 24.66% of the total. One possible explanation for this loss is that internal migrant workers are highly mobile and work shifts are too frequent during the two periods, making it difficult to track subjects. Meanwhile, a sample of 1146 internal migrant workers who stopped working as wage earners in 2014 was removed from the sample. This sample had accounted for 40.5 percent of the workers tracked in 2014. In order to test whether there are systematic differences between individuals in the lost sample as well as between individuals in the further deleted samples and those who do not fall into these two categories, we have added a balance test using the Probit model. As Table 2 in Appendix A shows, we generally did not find systematic differences between them.
Besides the missing individuals, the years considered in the research (2012-2014) are the years of the world economic crisis, which may affect the likelihood of migrant workers perceiving insufficient individual work demand [40-43] and experiencing salary reductions [44-46], which increase the presence of depressive symptoms. However, in China, the economic crisis was not as significant as in other parts of the world, especially for internal migrant workers from rural areas, because internal migrant workers can return to rural areas if they cannot find a suitable job in an urban area [47].
This study has a number of strengths. First, our sampling frame represents 95% of the Chinese population, and, therefore, can be considered nationally representative. Second, the large sample size of our panel data-set (n=1,686) gives our research a high degree of statistical power and considerable external validity. Third, the Institute of Social Science Survey (ISSS) of Peking University collected all data using a common sampling strategy. Last, this paper focuses on the relationship between depressive symptoms and internal migrant workers not only on their wage, but also on their job conversion and work duration. Some studies recognized that migrant workers are more likely to have occupational injuries [48-51]and may cause long absences and difficulty returning to work [52] which may induce a worsening of depressive symptoms. However, the conversion of occupation is one of the main factors recognized by WHO to reduce the burden of depression [53]. In this paper, we analyze the relationship between job conversion and wage decline.
Despite its strengths, our study also suffers from several limitations. First, the CFPS only collected self-reported information, which limited us to examining internal migrant workers’ depressive symptoms with medical evaluation. Second, because a person losing his or her job or earning less money probably is more likely to report depressive symptoms when interviewed with a survey, our research only indicates the association between internal migrant workers’ depressive symptoms and their income. Future research examining the causal effect of internal migrant workers’ depressive symptoms on their income in China would benefit from focusing on these research areas.
From a policy perspective, our results suggest that the universality of depressive symptoms in internal migrant workers must not be ignored. The Chinese government should publicize the existence of depressive symptoms and the harm they cause to internal migrant workers through various channels, so as to help them recognize their own potential depressive symptoms and lay a foundation for them to take further measures to address them. Relevant departments should fully consider the impact of unskilled human capital on policy effectiveness when formulating policies to improve human capital and achieve China's smooth crossing of the “middle income trap”. We believe our results can help policy makers develop targeted programs for depressed internal migrant workers in the future.
28. Al-Maskari, F.; Shah, S. M.; Al-Sharhan, R.; Al-Haj, E.; Al-Kaabi, K.; Khonji, D. , et al. Prevalence of depression and suicidal behaviors among male migrant workers in United Arab Emirates. J Immigr Minor Healt. 2011, 13(6), 1027-1032.
29. Lu, Y.; Hu, P.; Treiman, D. J. Migration and depressive symptoms in migrant-sending areas: findings from the survey of internal migration and health in China. Int J Public Health. 2012, 57(4), 691-698.
30. Mou, J.; Cheng, J.; Griffiths, S. M.; Wong, S. Y. S.; Hillier, S.; Zhang, D. Internal migration and depressive symptoms among migrant factory workers in Shenzhen, China. J Community Psychol. 2011, 39(2), 212-230.
31. El-Guebaly, N.; Currie, S.; Williams, J.; Wang, J. L.; Beck, C. A.; Maxwell, C.; Patten, SB. Association of mood, anxiety, and substance use disorders with occupational status and disability in a community sample. Psychiat. Serv. 2007, 58(5), 659-667.
32. Ettner, S. L.; Frank, R. G.; Kessler, R. C. The impact of psychiatric disorders on labor market outcomes. Nber Working Papers. 1997, 51(1), 64-81.
33. Karpansalo, M.; Kauhanen, J.; Lakka, T. A.; Manninen, P.; Kaplan, G. A.; Salonen, J. T. Depression and early retirement: prospective population based study in middle aged men. J. Epidemiol. Commun. H. 2005, 59(1), 70-74.
34. Lerner, D.; Adler, D. A.; Chang, H.; Lapitsky, L.; Hood, M. Y.; Perissinotto, C. et al. Unemployment, job retention, and productivity loss among employees with depression. Psychiat. Serv. 2004, 55(12), 1371-1378.
35. Dooley, D.; Prause, J.; Hamrowbottom, K. A. Underemployment and depression: Longitudinal relationships. J. Health Soc. Behav. 2000, 41 (4): 421-436.
36. Whooley, M. A.; Kiefe, C. I.; Chesney, M. A.; Markovitz, J. H.; Matthews, K.; Hulley, S. B. Depressive symptoms, unemployment, and loss of income: the cardia study. Arch Intern Med. 2008, 162(22), 2614-2620.
37. Dismuke, C. E.; Egede, L. E. Association between major depression, depressive symptoms and personal income in us adults with diabetes. Gen Hosp Psychiat. 2010, 32(5), 484-491.
38. Firdaus, G. Mental well-being of migrants in urban center of India: analyzing the role of social environment. Indian J Psychiat. 2017, 59(2), 164-169.
39. Robert, G.; Martínez, José Miguel; García, Ana M.; Benavides, F. G.; Ronda, E. From the boom to the crisis: changes in employment conditions of immigrants in Spain and their effects on mental health. Eur J Public Health. 2014, 24(3), 404-409.
40. Bergomi, M.; Modenese, A.; Ferretti, E.; Ferrari, A.; Licitra, G.; Vivoli, R. , et al. Work-related stress and role of personality in a sample of Italian bus drivers. Work. 2017, 57(3), 433-440.
41. Hoven, H.; Wahrendorf, M.; Siegrist, J. Occupational position, work stress and depressive symptoms: a pathway analysis of longitudinal share data. J Epidemiol Commun H. 2015, 69(5), 447-452.
42. Jurado D.; Gurpegui M.; Moreno O.; Fernández MC.; Luna JD.; Gálvez R. Association of personality and work conditions with depressive symptoms. Eur Psychiatry. 2005, 20(3), 213-222.
43. Mucci, N.; Giorgi, G.; Roncaioli, M.; Fiz Perez, J.; Arcangeli, G. The correlation between stress and economic crisis: a systematic review. Neuropsychiatr Dis Treat. 2016, 12, 983-993.
44. Chen, L.; Li, W.; He, J.; Wu, L.; Tang, W. Mental health, duration of unemployment, and coping strategy: a cross-sectional study of unemployed migrant workers in eastern China during the economic crisis. BMC Public Health. 2012, 12(1), 597.
45. Frasquilho, D.; Matos, M. G.; Salonna, F.; Guerreiro, D.; Storti, Cláudia C.; Gaspar, T. , et al. Mental health outcomes in times of economic recession: a systematic literature review. BMC Public Health. 2015, 16(1), 115.
46. Mattei, G.; Sacchi, V.; Alfieri, S.; Bisi, A.; Colombini, N.; Ferrari, S., et al. Stakeholders' views on vocational rehabilitation programs: a call for collaboration with occupational health physicians. Med Lav. 2018, 109(3), 201-209.
47. Chan, K. W. The global financial crisis and migrant workers in China: ‘there is no future as a labourer; returning to the village has no meaning’. Int J Urban Regional. 2010, 34(3), 659-677.
48. Ahonen, E.Q.; Benavides, F.G. Risk of fatal and non-fatal occupational injury in foreign workers in Spain. J Epidemiol Community Health. 2006, 60(5), 424-426.
49. Gobba, F.; Dall'Olio, E.; Modenese, A.; De Maria, M.; Campi, L.; Cavallini, GM. Work-related eye injuries: a relevant health problem. main epidemiological data from a highly-industrialized area of northern Italy. Int J Environ Res Public Health. 2017, 14(6), 604.
50. Salminen, S. Are Immigrants at Increased Risk of Occupational Injury? A Literature Review. The Ergonomics Open Journal. 2011, 4, 125-130.
51. Sabariego, C.; Coenen, M.; Ito, E.; Fheodoroff, K.; Scaratti, C.; Leonardi, M., et al. Effectiveness of integration and re-integration into work strategies for persons with chronic conditions: a systematic review of European strategies. Int J Environ Res Public Health. 2018, 15(3), 552.
52. Wisenthal, A.; Krupa, T. Cognitive work hardening: a return-to-work intervention for people with depression. Work. 2013, 45(4),423-430.
53. Organization, W. H. Mental health action plan 2013-2020. Lancet, 2013, 381(9882), 1970-1971.

Reviewer 2 Report
Dear Authors, Dear Editor,
The manuscript “Depressive Symptoms and Migrant Worker Wages: Estimation Based on a Nationally Representative Panel Dataset” discusses an interesting research and well designed research, but several major revisions have to be considered before the manuscript could be ready for a possible publication.
Keywords:
“China”: I think it is a too broad category for a keyword, maybe adding something before or after
“CEM”: I don’t know whether this abbreviation is so well known that it can be used as a keyword. I suggest to use the full word
Abstract:
- Line 10. Authors write “In recent years, Chinese migrant worker wage growth has slowed”. It is difficult for me to understand whether they are discussing of Chinese workers going abroad or of foreign workers going to China. I suggest a reformulation of this sentence.
INTRODUCTION
- Line 22. Authors don’t define who they identify for migrant workers. According to Wikipedia, a "migrant worker" is a person who either migrates within their home country or outside it to pursue work such as seasonal work. As I understand from the contraposition with “urban” worker, probably the Authors are using this second possibility for their definition of migrant workers. But there are many papers that adopt the definition of migrant workers as a synonymous for foreign workers coming to the studied Country. I suggest to reconsider the use of “migrant worker”, as too much tricky, or, if not, to state at the beginning of the paper what exactly is intended for “migrant workers” in the manuscript.
- Line 32: I am not an economist, but I am not sure China can be still considered a developing country.
- Line 39-42: Authors write “The literature already available suggests that negative psychological factors in young Americans have a negative impact on their future income. We are interested in whether the negative emotions of migrant workers have an impact on their wages”. This is too much generic for a scientific paper: please give a better definition of NEGATIVE (used 3 times!) emotions factors and impacts.
- Line 42: instead of “proportion of depressed”, I suggest prevalence of depressive symptoms.
- Lines 45-46: again, too much generic “impact of emotions”
- Line 52: not appropriate to speak about “effect”, it is only an association/correlation
- Line 54-57: please, have an accurate consideration of the use of the terms effectiveness, efficacy and efficiency that have different meanings
- Again, line 61 please avoid speaking of effects, only associations
- Section 2.1 . Authors are referring to the China Family Panel Studies (CFPS), and they report also the sampling technique for this research, but at least one, or better two, appropriate citation(s) reminding to these 2010 and 2014 researches from which the analysis of the paper have been performed are needed.
- Lines 90-97: in this part there is a little bit of confusions as the presentation of the possible resons why there are quite a lot of missing in the sample is a discussion of study limitations, and it should be moved to the discussion section. While the part introducing the characteristics of the sample reported in the Appendix stating that no statistical differences have been found is a presentation of a result of the study, and it has to be moved to the Results section. These two partes are both not appropriate for the materials and methods section.
- Lines 99-100. “The dependent variable in this study is migrant worker wages, measured by actual annual wages reported by the respondents”: more information is needed on this. How the question was formulated? Whathat does it mean “actual”? In what year?
- Line 102: better depressive symptoms instead of “symptoms of depression”
- Line 111: “Have there been job conversions between 2012 and 2014?”. This is not clear to me, English needs to be revised and this translation should be reconsidered
- Table 1. The description of the statistics is not required for variables as age, marital status, and completion of nine-year compulsory education: these variables only create confusion in the tabvle that results difficult to be read. Of course, if the sample is the same the age after two years is +2 years: data 2014 is not needed. For education there is no difference at all, and for the marital status the difference is very very small, so the description of the 2014 variables are not needed.
- Lines 124-137: I think that this part is not purely “methods”: it can be reduced moving something in the Introduction and something in the Discussion.
- Line 175: Better “relationships between depressive symptoms and wage” instead of “Effect of Depression”
- Line 176: Authors show the results of an analysis not presented in the method section!
- Table 2, Table 3 and Table 4: the tables need to be self-explicative. The title of the tables are unclear, and it is really difficult to understand what kind of results are presented in the table without an accurate reading of the text. Please change the table titles. And this can be applied also to Table 1.
- Line 222: again, effect: please avoid.
- Table 5: Please remove effect from the title, use “relationship” or “association”
- Discussion and references:
In the paper the discussion section is fully missing. A good international paper needs a discussion, with comparisons with results from other papers published also in other Countries, not only in China, possibly in very recent years, to cover the last updates in scientific literature. Currently the reference list includes only 30 citations, most of them are from Chinese papers, while the references citing non-Chinese papers are quite old. As indicated before, some parts of the introduction, and of the data/methods, and also of the results (in particular e.g. lines 230-239 where considerations on the number presented are discussed) sections can be used to write part of the discussion. Furthermore, in the discussion also a presentation of the study limitations has to be included, in particular the number of missing values for some variables, the subjective questionnaire based evaluation of depressive symptoms with no medical evaluation and consideration of depression risk factors has appropriate confounders to be inserted on the models, and the bias due to the fact that a person losing his job or earning less money probably is more likely to report depressive symptoms when interviewed with a survey.
Furthermore, new important points to be discussed with comparisons with other studies and citations of new and updated international references are:
1) The years considered in the research (2012-2014) are the years of the World economic crisis, that was not so important in China, compare to other parts of the world, but it may represent an important point to be considered, as it can affect the likelihood of migration of the workers, of loosing their jobs and reducing their salary and it increases the presence of depressive symptoms and other mental health problems and drugs/alcohol abuse, as reported in:
- Chen L, Li W, He J, Wu L, Yan Z, Tang W. Mental health, duration of unemployment, and coping strategy: a cross-sectional study of unemployed migrant workers in eastern China during the economic crisis. BMC Public Health. 2012 Aug 2;12:597. doi: 10.1186/1471-2458-12-597.
-Frasquilho D, Gaspar Matos M, Salonna F, et al: Mental health outcomes in times of economic recession: a systematic literature review. BMC Public Health 2016; 16: 115.
- Mattei G, Sacchi V, Alfieri S, Bisi A, Colombini N, Ferrari S, Giubbarelli G, Gobba F, Modenese A, Pingani L, Rigatelli M, Rossetti M, Venturi G, Starace F, Galeazzi GM. Stakeholders' views on vocational rehabilitation programs: a call for collaboration with Occupational Health Physicians. Med Lav. 2018 May 11;109(3):201-9. doi: 10.23749/mdl.v109i3.6844.
2) The possible role of work-related stress, that may arise not only from too much work but also from insufficient perceived individual work demand and it is associated with economic crisis, and it is strongly related to personality traits, including depressive symptoms, as shown by:
- Bergomi M, Modenese A, Ferretti E, Ferrari A, Licitra G, Vivoli R, Gobba F, Aggazzotti G. Work-related stress and role of personality in a sample of Italian bus drivers. Work. 2017;57(3):433-440. doi: 10.3233/WOR-172581.
- Hoven H, Wahrendorf M, Siegrist J. Occupational position, work stress and depressive symptoms: a pathway analysis of longitudinal SHARE data. J Epidemiol Community Health. 2015 May;69(5):447-52. doi: 10.1136/jech-2014-205206.
- Jurado D, Gurpegui M, Moreno O, Fernández MC, Luna JD, Gálvez R. Association of personality and work conditions with depressive symptoms. Eur Psychiatry. 2005 May;20(3):213-22.
- Mucci N, Giorgi G, Roncaioli M, Fiz Perez J, Arcangeli G. The correlation between stress and economic crisis: a systematic review. Neuropsychiatr Dis Treat. 2016 Apr 21;12:983-93. doi: 10.2147/NDT.S98525.
3) And finally I suggest to discuss also what can be a relevant strength of your study, that is the importance of understanding the associated factors influencing depression in migrant workers, as:
a) it is recognized that migrant workers are more likely to have occupational injuries and diseases as reported in:
- Ahonen E.Q., Benavides F.G., 2006. Risk of fatal and non-fatal occupational injury in foreign workers in Spain. J Epidemiol Community Health, 60(5):424-6. doi: 10.1136/jech.2005.044099
- Gobba F, Dall'Olio E, Modenese A, De Maria M, Campi L, Cavallini GM. Work-Related Eye Injuries: A Relevant Health Problem. Main Epidemiological Data from a Highly Industrialized Area of Northern Italy. Int J Environ Res Public Health. 2017 Jun 6;14(6). pii: E604. doi: 10.3390/ijerph14060604.
- Salminen S. Are Immigrants at Increased Risk of Occupational Injury? A Literature Review. The Ergonomics Open Journal, 2011, 4, 125-130. doi: 10.2174/1875934301104010125
AND
b) depressive symptoms are relevant causal factors associated with long-term work sickness absence, loosing of the job, and, more generally, with work-related illnesses. Moreover, relevant occupational diseases and injuries may frequent in migrant workers may also cause long-absence and difficulties in returning to work. The long absence and unemployment may induce a worsening of depressive symptoms, but the conservation of the occupation is one of the main factors recognized by WHO to reduce the burden of depression:
- Sabariego C, Coenen M, Ito E, Fheodoroff K, Scaratti C, Leonardi M, Vlachou A, Stavroussi P, Brecelj V, Kovačič DS, Esteban E. Effectiveness of Integration and Re-Integration into Work Strategies for Persons with Chronic Conditions: A Systematic Review of European Strategies. Int J Environ Res Public Health. 2018 Mar 19;15(3). pii: E552. doi: 10.3390/ijerph15030552.
- Wisenthal A, Krupa T. Cognitive work hardening: a return-to-work intervention for people with depression. Work. 2013;45(4):423-30. doi: 10.3233/WOR-131635.
- World Health Organization. Mental health action plan 2013 -2020 . https://apps.who.int/iris/bitstream/handle/10665/89966/9789241506021_eng.pdf;jsessionid=A7420F4F337A07448B660221CAD13C79?sequence=1
Lines 248-265: Conclusions
I’d like to read shorter and more effective conclusions. Some parts may be useful for the new discussion section. I’d suggest no more than 10-15 lines of conclusions.
Thank you and best regards,
The Reviewer
Author Response
Dear Qu:
Thank you very much for your letter, dated March 2, 2019. We want to thank you and the reviewer for your comments and suggestions on our manuscript, “Depressive Symptoms and Migrant Worker Wages: Estimation Based on a Nationally-Representative Panel Dataset” by Li Huang, Xue Zhang, Mi Zhou,Brendan Nuse and Liuyin Tong. We believe that the comments were helpful and have helped greatly improve the manuscript. Hence, we are sending you, as requested, the revised version of our manuscript.
In making our revisions we have responded to all suggestions by you and the reviewer. Generally, we made the following major revisions:
1.) We have added a discussion section. In this part, we discussed the main results, the interesting points, the strengths and limitations, and the policy implications of the study. We also compared our results with the results of other studies.
2.) We have changed the methodology section. We moved the mechanism to the discussion section. We also added an introduction to the OLS method.
3.) We have corrected a number of grammar and spelling mistakes.
4.) Following the suggestion of the reviewer, we have replaced the word “effect” with other words, such as “association” and “correlation”. We have also revised Table 1 (and related descriptions) accordingly.
We have included a detailed, point-by-point response to the reviewer in this letter. We are also providing a revised version of the manuscript in a Word document. We have tried to follow general IJERPH style, but, of course, stand ready to make any further changes that are needed (editorial or otherwise).
If you need anything else, please do not hesitate to contact me.
Sincerely,
Mi Zhou, Li Huang
Depressive Symptoms and Migrant Worker Wages: Estimation Based on a Nationally-Representative Panel Dataset
Response to Reviewer2:
First, we want to thank the reviewer for carefully reading our paper and providing useful, detailed comments – we believe addressing these comments has strengthened the paper considerably. We address each of the reviewer’s specific comments below. The reviewer’s comments are in italics. Our responses are indented.
Comment 1: “China”: I think it is a too broad category for a keyword, maybe adding something before or after.
“CEM”: I don’t know whether this abbreviation is so well known that it can be used as a keyword. I suggest to use the full word.
Response to Comment 1: This is a good point. In response, we have deleted the keyword China (line 19, page 1), and changed CEM to its full name, Coarsened Exact Matching method (lines 20-21, page 1):
“Keywords: migrant workers; depressive symptoms; wage; working stability; Coarsened Exact Matching method”
Comment 2: Line 10. Authors write “In recent years, Chinese migrant worker wage growth has slowed”. It is difficult for me to understand whether they are discussing of Chinese workers going abroad or of foreign workers going to China. I suggest a reformulation of this sentence.
Response to Comment 2: Thank you! To avoid confusion, we have changed this sentence to: Abstract: In recent years, migrant workers, people who move from Chinese rural areas to cities in other parts of the country to find work, have experienced slowed wage growth (lines 10-11, page 1). We have also added an explanation to the first sentence of the paper in order to make sure readers do not misunderstand our use of the term “migrant worker” (lines 24-26, page 1):
“Abstract: In recent years, migrant workers, people who move from Chinese rural areas to cities in other parts of the country to find work, have experienced slowed wage growth.”
“In recent years, the total number of migrant workers, which, in this paper, refers to workers who move from Chinese rural areas to cities in other parts of China to find work, has continuously increased.”
Comment 3: Line 22. Authors don’t define who they identify for migrant workers. According to Wikipedia, a "migrant worker" is a person who either migrates within their home country or outside it to pursue work such as seasonal work. As I understand from the contraposition with “urban” worker, probably the Authors are using this second possibility for their definition of migrant workers. But there are many papers that adopt the definition of migrant workers as a synonymous for foreign workers coming to the studied Country. I suggest to reconsider the use of “migrant worker”, as too much tricky, or, if not, to state at the beginning of the paper what exactly is intended for “migrant workers” in the manuscript.
Response to Comment 3: Thanks for pointing out this tricky phrase. We believe that the definition of the phrase added in the abstract and the first sentence of the paper eliminates confusion about our use of the term “migrant worker”.
Comment 4: Line 32: I am not an economist, but I am not sure China can be still considered a developing country.
Responses to Comments 4: Yes, China is still a developing country according to the studies of Li et al. (2017) and Nepelski et al. (2015).
Li.;Qing-Feng. The current challenge for plastic and reconstructive surgery in china, the biggest developing country. Journal of Craniofacial Surgery. 2017, 28(6), 1404.
Nepelski, D.; Prato, G. D. International technology sourcing between a developing country and the rest of the world. a case study of china. Technovation. 2015, 35, 12-21.
Comment 5: Line 39-42: Authors write “The literature already available suggests that negative psychological factors in young Americans have a negative impact on their future income. We are interested in whether the negative emotions of migrant workers have an impact on their wages”. This is too much generic for a scientific paper: please give a better definition of NEGATIVE (used 3 times!) emotions factors and impacts.
Response to Comment 5: We apologize for the generic expression and we have added a footnote to illustrate the meaning of negative psychological factors, negative impact and negative emotions (line 43, page 2).
“Negative emotions can be described as any feeling which causes you to be miserable and sad. These emotions make you dislike yourself and others, and take away your confidence.”
“Negative impact on wages means a reduction in earnings.
“Many of the negative psychological risk factors linked to dementia are associated with stress—or, more precisely, distress.”
Comment 6: Line 42: instead of “proportion of depressed”, I suggest prevalence of depressive symptoms.
Response to Comment 6: Thanks for your suggestion. We have changed “proportion of depressed” to “prevalence of depressive symptoms” (lines 45-46, page 2).
“More critically, according to estimates, the prevalence of depressive symptoms of migrant workers has reached 23.7% in China”
Comment 7: Lines 45-46: again, too much generic “impact of emotions”
Response to Comment 7: We completely agree. In response, we have changed “the impact” to “the association” (lines 48-49, page 2):
“Although there have been a number of studies examining the association between emotions and wages in China”
Comment 8: Line 52: not appropriate to speak about “effect”, it is only an association/correlation
Response to Comment 8: Yes. We have changed the effect to the association (lines 54-55, page 2).
“we examine the overall association between depressive symptoms and migrant worker wages by using a nationally representative panel dataset”
Comment 9: Line 54-57: please, have an accurate consideration of the use of the terms effectiveness, efficacy and efficiency that have different meanings
Response to Comment 9: The revised manuscript is as follows (lines 58-59, page 2):
“To perform this study, this paper uses a Coarsened Exact Matching (CEM) model, which can improve sample matching.”
Comment 10: Again, line 61 please avoid speaking of effects, only associations
Response to Comment 11: Thank you for catching our mistake. We have changed “effect” to “association”. The revised manuscript as follows (lines 63-64, page 2):
“It also examines the mechanisms of the association between depressive symptoms and migrant worker wages.”
Comment 11: Section 2.1 . Authors are referring to the China Family Panel Studies (CFPS), and they report also the sampling technique for this research, but at least one, or better two, appropriate citation(s) reminding to these 2010 and 2014 researches from which the analysis of the paper have been performed are needed.
Response to Comment 11: Thanks! We added two citations. The revised manuscript as follows (line 80, page 2):
“To create a nationally and provincially representative sample, the CFPS adopted a “Probability-Proportional-to-Size” (PPS) sampling strategy with multi-stage stratification and carried out a three-stage sampling process (Zhang et al.,2018; Zhou et al., 2018)”
Zhang, X.; Chen, X.; Zhang, X. The impact of exposure to air pollution on cognitive performance. Proceedings of the National Academy of Sciences. 2018, 115(37), 91-93.
Zhou, M.; Sun, X.; Huang, L.; Zhang, G.; Kenny, K.; Xue, H.; Auden, E.; Rozelle, S. Parental Migration and Left-Behind Children’s Depressive Symptoms: Estimation Based on a Nationally-Representative Panel Dataset. Int. J. Environ. Res. Public Health. 2018, 15, 1069-1082.
Comment 12: Lines 90-97: in this part there is a little bit of confusion as the presentation of the possible reasons why there are quite a lot of missing in the sample is a discussion of study limitations, and it should be moved to the discussion section. While the part introducing the characteristics of the sample reported in the Appendix stating that no statistical differences have been found is a presentation of a result of the study, and it has to be moved to the Results section. These two parties are both not appropriate for the materials and methods section.
Response to Comment 12: We apologize for the confusion. In our revised paper, we moved the explanation to the discussion section.
Comment 13: Lines 99-100. “The dependent variable in this study is migrant worker wages, measured by actual annual wages reported by the respondents”: more information is needed on this. How the question was formulated? Whathat does it mean “actual”? In what year?
Response to Comment 13: We apologize for the mistake. This has been corrected in the revised manuscript (line 92, page 3):
“The dependent variable in this study is migrant worker wages.”
Comment 14: Line 102: better depressive symptoms instead of “symptoms of depression”
Response to Comment 14: We apologize for the confusion. In the revised manuscript, we use depressive symptoms (line 94, page 3):
“The CES-D scale has been used widely in the international literature to evaluate depressive symptoms”
Comment 15: Line 111: “Have there been job conversions between 2012 and 2014?”. This is not clear to me, English needs to be revised and this translation should be reconsidered.
Response to Comment 15: We apologize for the confusion. We have changed this to “Did subjects change jobs between 2012 and 2014?” in the final manuscript (line 103, page 3).
“Did subjects change jobs between 2012 and 2014?”
Comment 16: Table 1. The description of the statistics is not required for variables as age, marital status, and completion of nine-year compulsory education: these variables only create confusion in the table that results difficult to be read. Of course, if the sample is the same the age after two years is +2 years: data 2014 is not needed. For education there is no difference at all, and for the marital status the difference is very very small, so the description of the 2014 variables are not needed.
Response to Comment 16: Thanks for your suggestion. We have deleted age and education in 2014. However, the age, marital status, and completion of nine-year compulsory education as explanation variables may be described to readers so that they know the distribution of our data. The revised table is now presented as follows (line 109, page 3):
Table 1. Descriptive statistics of variables.
Variables | Observations | Mean | SD | Min | Max |
Wage earnings in 2012(yuan) | 1686 | 21093.04 | 14236.94 | 79 | 80000 |
Wage earnings in 2014(yuan) | 1686 | 27546.04 | 17674.28 | 9 | 85000 |
Depressive symptoms in 2012(1 = yes, 0 = no) | 1686 | 0.29 | 0.45 | 0 | 1 |
Job conversions between 2012 and 2014(1 = yes, 0= no) | 1686 | 0.27 | 0.44 | 0 | 1 |
Work duration in 2012(months) | 919a | 19.83 | 7.38 | 0 | 33 |
Age in 2012 | 1686 | 35.94 | 10.66 | 18 | 60 |
Gender(1= Male, 0=Female) | 1686 | 0.70 | 0.46 | 0 | 1 |
Minority(1= Han Chinese, 0 = non-Han Chinese) | 1682b | 0.95 | 0.22 | 0 | 1 |
Marital status in 2012(1= Married, 0= others) | 1686 | 0.81 | 0.40 | 0 | 1 |
Marital status in 2014(1= Married, 0= others) | 1686 | 0.85 | 0.36 | 0 | 1 |
Completion of nine-year compulsory education in 2012 (1 = yes, 0 = no) | 1686 | 0.23 | 0.42 | 0 | 1 |
East area in 2012(1 = yes, 0= others) | 1686 | 0.45 | 0.50 | 0 | 1 |
Midland in 2012(1 = yes, 0= others) | 1686 | 0.34 | 0.47 | 0 | 1 |
East area in 2014(1 = yes, 0= others) | 1686 | 0.33 | 0.47 | 0 | 1 |
Midland in 2014(1 = yes, 0= others) | 1686 | 0.47 | 0.50 | 0 | 1 |
Sources: CFPS(2012, 2014)
Note: a The missing value comes from the fact that there was no work at the time of the survey in 2012, or the work duration data was missing at the time of the survey in 2014.
b Minority data has four missing values.
Comment 17: Lines 124-137: I think that this part is not purely “methods”: it can be reduced moving something in the Introduction and something in the Discussion.
Response to Comment 17: We apologize for the confusion. We have moved this part to the discussion section.
Comment 18: Line 175: Better “relationships between depressive symptoms and wage” instead of “Effect of Depression”
Response to Comment 18: We apologize for the confusion. In the revised manuscript, we have changed this phrase to “The relationship between depressive symptoms and wage”. The revised title is now presented as follows (line 161, page 5):
“4.1. The Relationship Between Depressive Symptoms and Migrant Worker Wage Earnings”
Comment 19: Line 176: Authors show the results of an analysis not presented in the method section!
Response to Comment 19: We apologize for the mistake. In the revised manuscript, we have added an introduction of the OLS method. The revised description is now presented as follows (lines 115-126, page 4):
“3.1 Ordinary Least squares Regression
Ordinary least-squares (OLS) regression is a generalized linear modelling technique that may be used to model a single response variable which has been recorded on at least an interval scale. The technique may be applied to single or multiple explanatory variables and also categorical explanatory variables that have been appropriately coded [23]
Ordinary Least squares Regression is represented as follows:
Where Y is the outcome variable, in this paper this refers to the wage.
X1 is a dummy variable to indicate whether the individual has depressive symptoms or not.
X2 represents other control variables, including gender, age, education level, marital status, and minority status.
β1, and β2 are regression coefficients and is the random error term.
Comment 20: Table 2, Table 3 and Table 4: the tables need to be self-explicative. The title of the tables are unclear, and it is really difficult to understand what kind of results are presented in the table without an accurate reading of the text. Please change the table titles. And this can be applied also to Table 1.
Response to Comment 20: We apologize for the confusion. The revised titles are as follows:
Table 2. Ordinary Least Squares estimates of the relationships between migrant worker depressive symptoms in 2012 and their wages in 2012 and 2014 (lines 171-172, page 5).
Table 3. Results of analysis conducted using Coarsened Exact Matching (line 187, page 6)
Table 4. Sample average treatment effect of migrant worker depressive symptoms in 2012 on their wages in 2014 (lines 194-195, page 6)
Comment 21: Line 222: again, effect: please avoid.
Response to Comment 21: We apologize for the confusion. In the revised manuscript, we have changed “effect” to “association”. The revised figure is now presented as follows (lines 208-209, page6):
“the association between depressive symptoms and internal migrant worker job conversion was significantly positive”
Comment 22: Table 5: Please remove effect from the title, use “relationship” or “association”
Response to Comment 22: Thanks for the suggestion. In the revised manuscript, we have replaced “effect” with “relationship” The revised figure is now presented as follows:
“Table 5. Relationship between depressive symptoms and employment stability in 2012. (lines 228, page8)”
Comment 23: Discussion and references:
In the paper the discussion section is fully missing. A good international paper needs a discussion, with comparisons with results from other papers published also in other Countries, not only in China, possibly in very recent years, to cover the last updates in scientific literature. Currently the reference list includes only 30 citations, most of them are from Chinese papers, while the references citing non-Chinese papers are quite old. As indicated before, some parts of the introduction, and of the data/methods, and also of the results (in particular e.g. lines 230-239 where considerations on the number presented are discussed) sections can be used to write part of the discussion. Furthermore, in the discussion also a presentation of the study limitations has to be included, in particular the number of missing values for some variables, the subjective questionnaire based evaluation of depressive symptoms with no medical evaluation and consideration of depression risk factors has appropriate confounders to be inserted on the models, and the bias due to the fact that a person losing his job or earning less money probably is more likely to report depressive symptoms when interviewed with a survey.
Furthermore, new important points to be discussed with comparisons with other studies and citations of new and updated international references are:
1) The years considered in the research (2012-2014) are the years of the World economic crisis, that was not so important in China, compare to other parts of the world, but it may represent an important point to be considered, as it can affect the likelihood of migration of the workers, of loosing their jobs and reducing their salary and it increases the presence of depressive symptoms and other mental health problems and drugs/alcohol abuse, as reported in:
- Chen L, Li W, He J, Wu L, Yan Z, Tang W. Mental health, duration of unemployment, and coping strategy: a cross-sectional study of unemployed migrant workers in eastern China during the economic crisis. BMC Public Health. 2012 Aug 2;12:597. doi: 10.1186/1471-2458-12-597.
-Frasquilho D, Gaspar Matos M, Salonna F, et al: Mental health outcomes in times of economic recession: a systematic literature review. BMC Public Health 2016; 16: 115.
- Mattei G, Sacchi V, Alfieri S, Bisi A, Colombini N, Ferrari S, Giubbarelli G, Gobba F, Modenese A, Pingani L, Rigatelli M, Rossetti M, Venturi G, Starace F, Galeazzi GM. Stakeholders' views on vocational rehabilitation programs: a call for collaboration with Occupational Health Physicians. Med Lav. 2018 May 11;109(3):201-9. doi: 10.23749/mdl.v109i3.6844.
2) The possible role of work-related stress, that may arise not only from too much work but also from insufficient perceived individual work demand and it is associated with economic crisis, and it is strongly related to personality traits, including depressive symptoms, as shown by:
- Bergomi M, Modenese A, Ferretti E, Ferrari A, Licitra G, Vivoli R, Gobba F, Aggazzotti G. Work-related stress and role of personality in a sample of Italian bus drivers. Work. 2017;57(3):433-440. doi: 10.3233/WOR-172581.
- Hoven H, Wahrendorf M, Siegrist J. Occupational position, work stress and depressive symptoms: a pathway analysis of longitudinal SHARE data. J Epidemiol Community Health. 2015 May;69(5):447-52. doi: 10.1136/jech-2014-205206.
- Jurado D, Gurpegui M, Moreno O, Fernández MC, Luna JD, Gálvez R. Association of personality and work conditions with depressive symptoms. Eur Psychiatry. 2005 May;20(3):213-22.
- Mucci N, Giorgi G, Roncaioli M, Fiz Perez J, Arcangeli G. The correlation between stress and economic crisis: a systematic review. Neuropsychiatr Dis Treat. 2016 Apr 21;12:983-93. doi: 10.2147/NDT.S98525.
3) And finally I suggest to discuss also what can be a relevant strength of your study, that is the importance of understanding the associated factors influencing depression in migrant workers, as:
a) it is recognized that migrant workers are more likely to have occupational injuries and diseases as reported in:
- Ahonen E.Q., Benavides F.G., 2006. Risk of fatal and non-fatal occupational injury in foreign workers in Spain. J Epidemiol Community Health, 60(5):424-6. doi: 10.1136/jech.2005.044099
- Gobba F, Dall'Olio E, Modenese A, De Maria M, Campi L, Cavallini GM. Work-Related Eye Injuries: A Relevant Health Problem. Main Epidemiological Data from a Highly Industrialized Area of Northern Italy. Int J Environ Res Public Health. 2017 Jun 6;14(6). pii: E604. doi: 10.3390/ijerph14060604.
- Salminen S. Are Immigrants at Increased Risk of Occupational Injury? A Literature Review. The Ergonomics Open Journal, 2011, 4, 125-130. doi: 10.2174/1875934301104010125
AND
b) depressive symptoms are relevant causal factors associated with long-term work sickness absence, loosing of the job, and, more generally, with work-related illnesses. Moreover, relevant occupational diseases and injuries may frequent in migrant workers may also cause long-absence and difficulties in returning to work. The long absence and unemployment may induce a worsening of depressive symptoms, but the conservation of the occupation is one of the main factors recognized by WHO to reduce the burden of depression:
- Sabariego C, Coenen M, Ito E, Fheodoroff K, Scaratti C, Leonardi M, Vlachou A, Stavroussi P, Brecelj V, Kovačič DS, Esteban E. Effectiveness of Integration and Re-Integration into Work Strategies for Persons with Chronic Conditions: A Systematic Review of European Strategies. Int J Environ Res Public Health. 2018 Mar 19;15(3). pii: E552. doi: 10.3390/ijerph15030552.
- Wisenthal A, Krupa T. Cognitive work hardening: a return-to-work intervention for people with depression. Work. 2013;45(4):423-30. doi: 10.3233/WOR-131635.
- World Health Organization. Mental health action plan 2013 -2020 . https://apps.who.int/iris/bitstream/handle/10665/89966/9789241506021_eng.pdf;jsessionid=A7420F4F337A07448B660221CAD13C79?sequence=1
Response to Comment 23: Thank you very much for the suggestion. In the revised manuscript, we have added a discussion section. The revised discussion is now presented as follows (lines 233-304, pages 8-9):
“5. Discussion
Using nationally representative data from the CFPS, we demonstrate that the prevalence of internal migrant worker depressive symptoms in China is high. Specifically, according to the CES-D scale, 29% of all internal migrant workers in China display symptoms of depression. These rates are high compared to similar studies that use nationally-representative samples from other developing countries [28]. These estimates also support previous studies in China that reported a high prevalence of internal migrant worker depression—although previous analyses used data from studies that were based on much smaller samples or from more limited geographic areas [29-30]. Taken together, these findings indicate that internal migrant workers’ depressive symptoms are indeed an issue in China, and steps must be taken to improve the mental health of Chinese internal migrant workers.
Our results also show that internal migrant workers with depressive symptoms were more prone to switch jobs and lacked job stability. In addition, we discovered that internal migrant workers with depressive symptoms stayed at the same job for a shorter amount of time than those without depressive symptoms did. As many studies have shown, the unemployment rate and probability of unemployment are significantly higher for those with depressive symptoms than for those without depressive symptoms [31-34]. At the same time, depression may also prolong the gap period between jobs [35]. These findings are important because they have allowed us to identify the mechanism determination of internal migrant worker wages in China. With this knowledge, it will be possible to develop policies and programs targeting these internal migrant workers’ depressive symptoms that can increase their productivity across China.
Although no previous studies have comprehensively examined the relationship between depressive symptoms and internal migrant worker wages in China, our findings are supported by the international literature. For example, a study conducted in the United States found that depressive symptoms are associated with subsequent unemployment and loss of family income among working young adults [36]. In addition, another study from the United States found that major depression was associated with $2,838 lower personal income [37]. Findings from other countries on the relationship between depressive symptoms and income also support the findings of our research. Research conducted in India has found that mental health was positively correlated with higher income level [38]. It has also been found in a Spanish sample that migrant workers whose monthly income decreased suffered an increased risk of poor mental health [39].
We use panel data to test whether depressive symptoms are associated with wage. But there are 927 individuals missing from 2012 to 2014, accounting for 24.66% of the total. One possible explanation for this loss is that internal migrant workers are highly mobile and work shifts are too frequent during the two periods, making it difficult to track subjects. Meanwhile, a sample of 1146 internal migrant workers who stopped working as wage earners in 2014 was removed from the sample. This sample had accounted for 40.5 percent of the workers tracked in 2014. In order to test whether there are systematic differences between individuals in the lost sample as well as between individuals in the further deleted samples and those who do not fall into these two categories, we have added a balance test using the Probit model. As Table 2 in Appendix A shows, we generally did not find systematic differences between them.
Besides the missing individuals, the years considered in the research (2012-2014) are the years of the world economic crisis, which may affect the likelihood of migrant workers perceiving insufficient individual work demand [40-43] and experiencing salary reductions [44-46], which increase the presence of depressive symptoms. However, in China, the economic crisis was not as significant as in other parts of the world, especially for internal migrant workers from rural areas, because internal migrant workers can return to rural areas if they cannot find a suitable job in an urban area [47].
This study has a number of strengths. First, our sampling frame represents 95% of the Chinese population, and, therefore, can be considered nationally representative. Second, the large sample size of our panel data-set (n=1,686) gives our research a high degree of statistical power and considerable external validity. Third, the Institute of Social Science Survey (ISSS) of Peking University collected all data using a common sampling strategy. Last, this paper focuses on the relationship between depressive symptoms and internal migrant workers not only on their wage, but also on their job conversion and work duration. Some studies recognized that migrant workers are more likely to have occupational injuries[48-51]and may cause long absences and difficulty returning to work [52] which may induce a worsening of depressive symptoms. However, the conversion of occupation is one of the main factors recognized by WHO to reduce the burden of depression [53]. In this paper, we analyze the relationship between job conversion and wage decline.
Despite its strengths, our study also suffers from several limitations. First, the CFPS only collected self-reported information, which limited us to examining internal migrant workers’ depressive symptoms with medical evaluation. Second, because a person losing his or her job or earning less money probably is more likely to report depressive symptoms when interviewed with a survey, our research only indicates the association between internal migrant workers’ depressive symptoms and their income. Future research examining the causal effect of internal migrant workers’ depressive symptoms on their income in China would benefit from focusing on these research areas.
From a policy perspective, our results suggest that the universality of depressive symptoms in internal migrant workers must not be ignored. The Chinese government should publicize the existence of depressive symptoms and the harm they cause to internal migrant workers through various channels, so as to help them recognize their own potential depressive symptoms and lay a foundation for them to take further measures to address them. Relevant departments should fully consider the impact of unskilled human capital on policy effectiveness when formulating policies to improve human capital and achieve China's smooth crossing of the “middle income trap”. We believe our results can help policy makers develop targeted programs for depressed internal migrant workers in the future.
28. Al-Maskari, F.; Shah, S. M.; Al-Sharhan, R.; Al-Haj, E.; Al-Kaabi, K.; Khonji, D. , et al. Prevalence of depression and suicidal behaviors among male migrant workers in United Arab Emirates. J Immigr Minor Healt. 2011, 13(6), 1027-1032.
29. Lu, Y.; Hu, P.; Treiman, D. J. Migration and depressive symptoms in migrant-sending areas: findings from the survey of internal migration and health in China. Int J Public Health. 2012, 57(4), 691-698.
30. Mou, J.; Cheng, J.; Griffiths, S. M.; Wong, S. Y. S.; Hillier, S.; Zhang, D. Internal migration and depressive symptoms among migrant factory workers in Shenzhen, China. J Community Psychol. 2011, 39(2), 212-230.
31. El-Guebaly, N.; Currie, S.; Williams, J.; Wang, J. L.; Beck, C. A.; Maxwell, C.; Patten, SB. Association of mood, anxiety, and substance use disorders with occupational status and disability in a community sample. Psychiat. Serv. 2007, 58(5), 659-667.
32. Ettner, S. L.; Frank, R. G.; Kessler, R. C. The impact of psychiatric disorders on labor market outcomes. Nber Working Papers. 1997, 51(1), 64-81.
33. Karpansalo, M.; Kauhanen, J.; Lakka, T. A.; Manninen, P.; Kaplan, G. A.; Salonen, J. T. Depression and early retirement: prospective population based study in middle aged men. J. Epidemiol. Commun. H. 2005, 59(1), 70-74.
34. Lerner, D.; Adler, D. A.; Chang, H.; Lapitsky, L.; Hood, M. Y.; Perissinotto, C. et al. Unemployment, job retention, and productivity loss among employees with depression. Psychiat. Serv. 2004, 55(12), 1371-1378.
35. Dooley, D.; Prause, J.; Hamrowbottom, K. A. Underemployment and depression: Longitudinal relationships. J. Health Soc. Behav. 2000, 41 (4): 421-436.
36. Whooley, M. A.; Kiefe, C. I.; Chesney, M. A.; Markovitz, J. H.; Matthews, K.; Hulley, S. B. Depressive symptoms, unemployment, and loss of income: the cardia study. Arch Intern Med. 2008, 162(22), 2614-2620.
37. Dismuke, C. E.; Egede, L. E. Association between major depression, depressive symptoms and personal income in us adults with diabetes. Gen Hosp Psychiat. 2010, 32(5), 484-491.
38. Firdaus, G. Mental well-being of migrants in urban center of India: analyzing the role of social environment. Indian J Psychiat. 2017, 59(2), 164-169.
39. Robert, G.; Martínez, José Miguel; García, Ana M.; Benavides, F. G.; Ronda, E. From the boom to the crisis: changes in employment conditions of immigrants in Spain and their effects on mental health. Eur J Public Health. 2014, 24(3), 404-409.
40. Bergomi, M.; Modenese, A.; Ferretti, E.; Ferrari, A.; Licitra, G.; Vivoli, R. , et al. Work-related stress and role of personality in a sample of Italian bus drivers. Work. 2017, 57(3), 433-440.
41. Hoven, H.; Wahrendorf, M.; Siegrist, J. Occupational position, work stress and depressive symptoms: a pathway analysis of longitudinal share data. J Epidemiol Commun H. 2015, 69(5), 447-452.
42. Jurado D.; Gurpegui M.; Moreno O.; Fernández MC.; Luna JD.; Gálvez R. Association of personality and work conditions with depressive symptoms. Eur Psychiatry. 2005, 20(3), 213-222.
43. Mucci, N.; Giorgi, G.; Roncaioli, M.; Fiz Perez, J.; Arcangeli, G. The correlation between stress and economic crisis: a systematic review. Neuropsychiatr Dis Treat. 2016, 12, 983-993.
44. Chen, L.; Li, W.; He, J.; Wu, L.; Tang, W. Mental health, duration of unemployment, and coping strategy: a cross-sectional study of unemployed migrant workers in eastern China during the economic crisis. BMC Public Health. 2012, 12(1), 597.
45. Frasquilho, D.; Matos, M. G.; Salonna, F.; Guerreiro, D.; Storti, Cláudia C.; Gaspar, T. , et al. Mental health outcomes in times of economic recession: a systematic literature review. BMC Public Health. 2015, 16(1), 115.
46. Mattei, G.; Sacchi, V.; Alfieri, S.; Bisi, A.; Colombini, N.; Ferrari, S., et al. Stakeholders' views on vocational rehabilitation programs: a call for collaboration with occupational health physicians. Med Lav. 2018, 109(3), 201-209.
47. Chan, K. W. The global financial crisis and migrant workers in China: ‘there is no future as a labourer; returning to the village has no meaning’. Int J Urban Regional. 2010, 34(3), 659-677.
48. Ahonen, E.Q.; Benavides, F.G. Risk of fatal and non-fatal occupational injury in foreign workers in Spain. J Epidemiol Community Health. 2006, 60(5), 424-426.
49. Gobba, F.; Dall'Olio, E.; Modenese, A.; De Maria, M.; Campi, L.; Cavallini, GM. Work-related eye injuries: a relevant health problem. main epidemiological data from a highly-industrialized area of northern Italy. Int J Environ Res Public Health. 2017, 14(6), 604.
50. Salminen, S. Are Immigrants at Increased Risk of Occupational Injury? A Literature Review. The Ergonomics Open Journal. 2011, 4, 125-130.
51. Sabariego, C.; Coenen, M.; Ito, E.; Fheodoroff, K.; Scaratti, C.; Leonardi, M., et al. Effectiveness of integration and re-integration into work strategies for persons with chronic conditions: a systematic review of European strategies. Int J Environ Res Public Health. 2018, 15(3), 552.
52. Wisenthal, A.; Krupa, T. Cognitive work hardening: a return-to-work intervention for people with depression. Work. 2013, 45(4),423-430.
53. Organization, W. H. Mental health action plan 2013-2020. Lancet, 2013, 381(9882), 1970-1971.
Comment 24: Lines 248-265: Conclusions
I’d like to read shorter and more effective conclusions. Some parts may be useful for the new discussion section. I’d suggest no more than 10-15 lines of conclusions.
Response to Comment 24: Thanks for the suggestion. In the revised manuscript, we kept only the 1st paragraph of the conclusion. The revised conclusion is now presented as follows (lines 306-313, page 9):
“Using nationally representative data, this study explored the relationship between depressive symptoms and internal migrant worker wages. In addition, we examined the mechanism between depressive symptoms and internal migrant worker wages from the perspectives of job conversion and work duration. Our results showed that nearly 30% of internal migrant workers exhibited depressive symptoms, which had a significant direct negative consistent relationship with internal migrant worker wages. Furthermore, depressive symptoms had a significant indirect relationship with internal migrant worker wages through job stability. In particular, depressive symptoms can reduce internal migrant worker wages by increasing the frequency of job conversion or shortening the duration of the work.”

Round 2
Reviewer 2 Report
Dear Editor and Authors of the Manuscript "Depressive Symptoms and Migrant Worker Wages: Estimation Based on a Nationally-Representative Panel Dataset",
congratulations! I am more than happy with the revisions provided that appropriately addressed all my comments and suggestions. I think that the paper has been significantly improved and now it could be ready for publication in IJERPH.
Best regards,
the Reviewer